# Automatic Placement of Visual Sensors in a Smart Space to Ensure Required PPM Level in Specified Regions of Interest

**DOI:** 10.3390/s22207806

**Published:** 2022-10-14

**Authors:** Iaroslav Khutornoi, Aleksandr Kobyzhev, Irina Vatamaniuk

**Affiliations:** Laboratory “Industrial Systems for Streaming Data Processing” of SPbPU NTI Center, Peter the Great St. Petersburg Polytechnic University, Polytechnicheskaya, 29, 195251 Saint Petersburg, Russia

**Keywords:** camera placement, surveillance, pinhole camera, highlights, multi-objective optimization, genetic algorithm, topsis

## Abstract

This work is devoted to a cost-effective method for the automatic placement of visual sensors within a smart room to ensure the requirements for its design. Various unique conditions make the process of manually placing sensors time consuming and can also lead to a decrease in system efficiency. To automate the design process, we solve a multi-objective optimization problem known as the art gallery problem in 3D, modified as follows. For the specified regions of interest within a smart room, the required pixels per meter level (PPM) should be ensured. The optimization criteria are visibility of the room and the cost of equipment. To meet these criteria, we describe a room model with doors, windows, and obstacles represented in such a way as to consider their impact on the field of view of the sensors. To model sensor placement, a genetic algorithm is used. The optimal solution is selected from the Pareto front by means of the technique for order of preference by similarity to ideal solution (TOPSIS). The developed method’s effectiveness has been tested on modeling real premises of various types. The method is flexible because of the assignment of weights to certain aspects when placing sensors. Further, it can be scalable to other types of sensors.

## 1. Introduction

Today, the high level of automation in different areas requires equipping premises with a variety of sensors and devices for effective production of goods and service provision (automated production lines, intelligent meeting rooms, stores with no cashiers, etc.). Sensor placement in premises largely depends on the particular solutions of specialists who rely on their own experience and perceptions of benefit. Their solutions are usually acceptable for basic monitoring and surveillance. However, when it comes to equipping premises for more specific purposes, the task of sensor placement becomes more complicated and harder to scale. For instance, it is often difficult to choose the acceptable configuration of visual sensors in big premises due to various unique conditions that need to be taken into account at specific facilities, e.g., natural light sources (if any), obstacles, door arrangement, etc. These aspects make the manual design of sensor placement time-consuming, and the chosen solution might not be the best one. Thus, an urgent problem consists in the elaboration of an automatic sensor placement method, which will reduce the design error occurrence, as well as design time and cost. In addition, budget constraints, room features, requirements for sensor types, as well as requirements for signal characteristics should be taken into account.

In fact, visual sensors’ automatic placement is a particular case of solving the well-known art gallery problem (AGP) in three-dimensional space with a given set of constraints [1]. It can be stated as follows. The task is to equip a three-dimensional room of any geometric shape with the smallest number of sensors with a given range and resolution to ensure the room’s full coverage, taking into account the solution cost and image quality requirements (if any).

In this work, we consider the equipment of a smart room with visual sensors to provide: (1) full visibility of the room; (2) required pixels per meter (PPM) level in given regions of interest (RoI); and (3) the lowest possible cost of equipment. Further in the paper, we discuss closed-circuit television (CCTV) camera placement, although the type of sensor can be any (infrared, multispectral cameras, audio sensors, etc.)

The solution proposed in this paper is based on the use of a 3D model of the room, which makes it possible to calculate the exact view area of each of the visual sensors in the scene, taking into account all significant obstacles and complex geometry of the room. Applying the proposed solution, it is possible to unambiguously determine whether a person enters full-length into the frame and whether any region of interest is fully visible with the required image quality. In addition, the proposed algorithm allows taking into consideration the vertical angle between the camera and the door, which provides the best image quality for recognizing and identifying a person entering the room. Furthermore, the option to project scene objects into the camera image plane enables determining the window light and taking it into account when placing cameras in the room. All these aspects allow calculating the cameras’ view area so as to take into account the image requirements according to the premises specifics. As well, the proposed method can be scaled for other types of sensors and devices to ensure full equipment of intelligent indoor systems.

## 2. Related Works

Let us consider several modern approaches to solving Art Gallery Problem (Table 1). Horster and Lienhart [2] proposed an algorithm for placing visual sensors with different characteristics in a two-dimensional rectangular area while minimizing the cost of the equipment used. This approach involves the use of cameras with different fields of view and costs. Using the intrinsic parameters of the camera, the field of view of the sensor is calculated as a triangular polygon on the floor plane. The sensor installation area is discretized, forming a grid of points where cameras can be placed. To solve this problem, the integer programming model is used. The algorithm results in the minimum number of cameras with the minimum total cost that maximize the coverage of a given rectangular area.

Abdalla and Asirvadam [3] proposed a method of automatic visual sensor placement, in which the scene is represented as a 2D floor-plan, and the camera field of view can be represented by one of four figures on the floor plane: trapezoids, triangles, and their analogues with radial bases. When calculating the camera field of view, dead zones caused by the walls of the room are taken into consideration. The algorithm results in the room coverage (%), considering area significance in the room. Comparing the methods of representing the camera field of view on the floor plane, the authors identified a trapezoid and a triangle without radial bases as the most effective.

Van den Hengel and Hill [4] described a cellular decomposition-based method for determining the best placement option of a large number of cameras in any building layouts. The method takes a 3D model of a building as input and uses a genetic algorithm to find a location that maximizes coverage and minimizes the cameras’ overlap. The room floor is discretized, forming a grid of cells that should be covered with visual sensors. To generate camera positions, a correction algorithm is used. It transfers a point inside the room as if it was generated outside of it. When calculating the algorithm metric, both the coverage of cells and the number of their overlaps by different cameras in the room are taken into account.

Kim and Ham [5] proposed a camera placement algorithm for the use in the complex facilities construction. The workspace is divided into cells for which weights are set. This allows placing cameras so as to cover crucial points (e.g., work areas or areas of people’s movement) with a limited number of cameras. The algorithm considers the height of camera placement, but in fact, all calculations are carried out in 2D plane. The authors discuss maximizing the total coverage of cameras, taking into account the weights of different areas, as well as minimizing the costs associated with buying and installing cameras. The installation points are determined by three conditions of the work site: the facility boundaries, the power supply availability, and the data transmission possibility. The task of multi-objective optimization is solved by the genetic algorithm.

Sourav and Peschel [6] have developed a method for finding the best points to place visual sensors inside large cowsheds to monitor the animals behavior and physical condition. The proposed method uses a voxelized 3D scene representation, visibility cone camera model, and ray tracing to find dead zones near objects. The task of multi-objective optimization is solved by the genetic algorithm, which receives two objective functions and imposed restrictions as input. The first function maximizes camera coverage based on priority coverage areas based on weighted voxels. The second function minimizes the cost of the camera, its installation, and maintenance. The study found that the optimal location of cameras can be determined during the first few iterations of the genetic algorithm. The authors note that maximizing coverage by the camera’s angle of view is of utmost importance.

Amiri and Rohani [7] have developed a decision support system for the CCTV camera installation using the advanced hybrid particle swarm optimization decision support system (HPSO-DSS) particle swarm optimization algorithm. The system operates the cameras installation height and tilt when calculating its view area. Further, the algorithm takes into account dead zones caused by walls. However, objects in the room are not considered when calculating the view area. When calculating dead zones, the upper and lower bound height of the camera’s view area are taken into account, which makes it possible to set a three-dimensional space that should be observed. The algorithm aims to maximize both the entire room coverage and the regions of interest with a given PPM. If a door falls into the camera’s view area, the camera must be turned to the door at a certain horizontal angle, otherwise the coverage of the door zone will not be taken into consideration when calculating the objective function value.

Albahri and Hammad [8] proposed a method involving the use of building information modeling (BIM) to optimize the number and location of CCTV cameras. BIM is applied to determine the input data of the optimization process and to visualize the results. The main feature of the proposed method is the division of the room into a 2D cell grid, which is used to facilitate the placement of cameras within the valid area. Furthermore, the space of the room is divided into areas which are assigned a certain importance value. This allows focusing on certain areas of the room (e.g., elevators or escalators) when placing cameras automatically. To optimize the placement of cameras, the genetic algorithm is also used. The findings indicate that the maximum coverage of the room can be achieved by placing cameras with different angles of view near the edge of the workspace with the appropriate yaw and pitch angles.

Thus, the use of a 3D model is more advantageous for the automatic placement of cameras on a complex scene because it allows for taking into account the height of obstacles and the vertical angle relative to objects. A significant limitation when calculating the camera’s view area is the upper bound height, which provides an opportunity to reliably observe objects below a given height. As for the room model, it should take into account the dead zones formed by stationary obstacles. Another important component is the room regions of interest, which must be covered first of all when placing cameras. The task of automatic camera placement should be multi-objective, i.e., it should consist of both finding the optimal cameras installation positions and minimizing the cameras cost. Objective functions should have limitations on the camera placement area and on the available budget.

Within the techniques considered, the approach proposed by Albahri and Hammad [8] takes into account most of the features of the room. However, it makes no calculation of the angle between the camera and the door, as well as the budget limit. Moreover, none of the considered techniques take into account the window light, which is one of the most significant features of the room when placing cameras indoors. High-contrast lighting conditions make the tasks of human and object recognition barely solvable and require more expensive equipment (e.g., wide dynamic range (WDR) cameras).

Let us also consider some studies devoted to optimization algorithms in the problem of automatic placement of sensors. A thorough comparison of mainstream coverage optimization methods is presented by Xuebo Zhang and Boyu Zhang in [9]. The authors compare the greedy algorithm, genetic algorithm (GA), particle swarm optimization (PSO), binary integer programming (BIP), and differential evolution algorithm (DE) in solving typical camera deployment problems for the coverage of 3D objects. The coverage metric is the number of polygons in the view area of the cameras. Authors show that the GA provides better computational efficiency while lacking coverage performance. Opposite to it, the BIP has the best results among the considered methods while being the most time consuming. Further, the authors present several approaches to improve the computational efficiency, which can be useful for implementing finite software.

Rangel and Costa [10] formalize the multi-objective redundant coverage maximization problem in Wireless Visual Sensors Networks (WVSN) and propose two evolutionary algorithms for solving it. To ensure coverage redundancy, they specify several target criteria: (1) maximization of the coverage of target objects by visual sensors; (2) maximization of redundancy coverage of objects, in which each object is observed by two or more visual sensors; and (3) maximization of the average number of visual sensors used for redundant coverage of one object. The authors propose an a priori lexicographic algorithm and a posterior NSGA-II algorithm and discuss their benefits. Thus, the authors show that evolutionary algorithms can effectively solve multi-criteria problems, while offering a choice from a variety of solutions.

Kovacs and Bolemanyi [11] consider the task of terrain large scale optical sensor placement. To optimize the sensor placement, the authors propose to find targets at one point of their possible route, and not to cover the entire territory with sensors. The environment model consists of terrain, clouds, vegetation, and artificial building elements, for which visibility degradation and background attenuation of the object are taken into account. Optimization was performed in 3D using a bacterial evolutionary algorithm which took into account the various features of all considered objects. It is interesting to note that the proposed approach does not focus on the full coverage of the area under consideration but gives the satisfactory result by the competent picking of the sensor placement points.

Thus, the considered studies show that evolutionary algorithms are suitable for multicriteria optimization of automatic placement of sensors. With their help, the most proper position of the sensors can be selected, which is rather difficult for classical methods in conditions of a large number of parameters. It is also worth noting that it is important to correctly apply optimization approaches to every specific task. Therefore, the genetic algorithm can form the Pareto front at the output, on which the optimal solutions are found. This allows us to choose the best solution according to a certain metric.

## 3. Materials and Methods

Most of the reviewed studies consider all calculations in a 2D scene. This leads to a decrease in the accuracy of determining the camera view area due to the inability to take into account the three-dimensional features of obstacles, doors, and windows. Taking into account the height of obstacles in 2D mode is applicable only for simple-shaped objects, whereas it is impossible to consider all the features of objects of complex shape in 2D scene.

Therefore, we present the scene model as a three-dimensional space allowing for the following elements: obstacles for the dead zones analysis; regions of interest (RoI) with a required PPM level to be ensured for crucial points and object monitoring; doors are an important element of the scene, for which it is necessary to take into account both the PPM and the angle of view; and windows should be considered in order to prevent possible highlights and flares in the image. The scene representation in such a way provides an opportunity to simulate a real room with high accuracy and is sufficient to solve the problem of automatic visual sensor placement.

The sensor model is represented by a projective camera model, which allows calculating the exact view area in a 3D environment with specified image quality levels in PPM, taking into account dead zones caused by obstacles and high-contrast lighting from windows. The view area is calculated considering the desired upper bound visibility height, then the obtained field of view is projected onto the floor plane. All subsequent operations are carried out with the view area polygons in 2D mode, which simplifies the calculations and improves performance (pseudo 3D).

Using the models of the room and the camera described above, the visual sensor placement is simulated by a genetic algorithm. Then, the room coverage metrics are calculated, and the optimal sensor installation points are selected (maximizing the room coverage while minimizing the cost of the equipment used). Whereas the problem is multi-objective, we consider a set of optimal Pareto solutions, then pick the most appropriate of them via the criteria set. The scheme of the proposed method is shown in Figure 1.

Let us consider the proposed method of automatic placement of visual sensors in detail.

### 3.1. Room Model

The room model is represented as a set of triangulated objects defined by a set of vertices and faces, and includes floor, ceiling, walls, doors, windows, stationary objects (obstacles), and regions of interest (Figure 2). Moreover, the model can contain a specified allowed area for placing cameras, which is taken into account when placing them automatically.

Regions of interest can be presented as polygons on any plane in the room, which must be covered by the view area of one of the cameras with the minimum required PPM level, as shown in Figure 2. The region of interest is commonly located on the floor plane next to some important objects of the scene (e.g., a security checkpoint at the organization entrance, the speaker’s location area in the conference room, etc.). When the RoI is located on the floor, its visibility is guaranteed with the required minimum PPM level, taking into account the pre-determined desired upper bound visibility height. Thus, we can guarantee that the visibility level acceptable for the RoI purposes will be provided in the camera’s field of view from the floor to a given height. The following values of the desired PPM levels are accepted in the model (Table 2).

The obstacles can be of two types: objects forming the cameras’ dead zones (pink) and ghost obstacles which are used only for visualization on the scene (light blue). The classification of obstacles allows the consideration of only those objects that affect the placement of cameras in the room (e.g., cabinets, partitions, and columns). These objects are either always or often stationary and occupy a large amount of space, which is why dead zones of considerable size are formed. Other objects (e.g., tables and chairs) are not important when considering the dead zones because their height is quite low, or their location changes frequently. These objects are intended only for the most complete representation of the room when visualizing it.

The doors are represented as a following set of features: a flat triangulated three-dimensional rectangular object in the wall; a RoI on the floor plane; the door opening direction; and the door handle (Figure 2). The RoI of the door sets the area next to the door in which a person is guaranteed to be seen in full height when entering or exiting the room, taking into account the required visibility height. The position of the door handle is considered in order to minimize the likelihood of the dead zone appearance, caused by the door opened inside the room. By minimizing both the horizontal and vertical angle relative to the door’s normal, we achieve a more convenient angle of view, thereby increasing the probability of various objects in the doorway (primarily people), in terms of recognition and detection.

The doors are also divided into two classes: (1) main doors, for which the angles relative to the camera are taken into account (gray in Figure 2); and (2) secondary doors, for which the angles relative to the camera are not considered (orange in Figure 2). This classification allows taking into account the entrance doors that should be covered first because it is through them that a potentially new person can enter the room. The secondary doors are considered to have no exit outside, i.e., a person who enters them is guaranteed to return to the room from the same or another secondary door. For this class of doors, the horizontal and vertical angle relative to the camera is not taken into account because the task is to ensure that the door is in the visual sensor view area like any other object in the room.

### 3.2. Visual Sensor Model

The visual sensor model is represented by a mathematical model of a projective camera (pinhole camera), which describes the transformation of the world coordinates of a point into the coordinates of a point in the image:(1)m=K⋅[R|t]⋅M,
where *K* is the matrix of the camera intrinsic parameters, *R* is the rotation matrix of the camera, *t* is the translation vector of the camera (| denotes concatenation of the matrix *R* and the vector *t*), *M* is the coordinates of a point in three-dimensional space, and *m* is the coordinates of a point in the image.

The camera-intrinsic parameters describe the camera projection model properties, which transform the world coordinates relative to the camera to the coordinates in the image. The extrinsic parameters of the camera *R*, *t* describe the position and spatial orientation of the camera relative to the observed scene.

When calculating the camera’s field of view, the near and far clipping planes are used parallel to the projective plane that defines the visible part of the scene.

To determine the camera’s field of view, the intersection line of the camera’s far clipping plane and the scene plane is calculated. The resulting intersection line is a type of a horizon line relative to a given plane, beyond which the visibility lacks. After this, the intersection line is projected onto the camera plane according to Equation (1). Then, the clipping side of the invisible part of the scene is determined by projecting the normal vector of the selected plane onto the image. The direction of this vector determines the invisible side. The visualization of the projected line on the camera plane relative to the floor plane is shown in Figure 3a.

Thus, it is possible to project the camera’s field of view with a given PPM level onto the selected scene plane by setting the distance of the far clipping plane to a certain value. The distance to the far clipping plane corresponding to the specified PPM level is calculated based on the camera parameters:(2)$$D=\frac{I_{w}}{2\cdot ppm\cdot tan\left(\frac{fov_{h}}{360\cdot \pi }\right)},$$
where $I_{w}$ is the width of the image in pixels; ${\mathrm{fov}}_{h}$ is the horizontal angle of view of the camera; and *ppm* is the camera resolution in pixels (Table 2).

After calculating the camera’s field of view in the image plane, we can project it onto the 3D plane via a reverse projection, which allows us to convert the coordinates of a point in the image into world coordinates:(3)$$\left[\begin{array}{c} X\\ Y\\ Z \end{array}\right]=k\cdot R\cdot K^{-1}\cdot \left[\begin{array}{c} u\\ v\\ 1 \end{array}\right]+t.$$

Thus, we get the coordinates of the camera’s visibility polygon on any plane of the scene with the required PPM level.

The calculation of the camera’s view area, taking into account the dead zones caused by room walls and obstacles, is performed by pre-projecting a 3D object on the camera image to determine its visible part and its subsequent reverse projection onto the selected scene plane.

The objects of the scene are pre-cut in 3D by the near camera clipping plane and the floor plane. The remaining (visible) part of the object is projected onto the image plane of the camera to limit the far edge of visibility and the boundaries of the frame (the viewing sector), after which the external contours (polygons) of the objects are determined. The obstacle polygons obtained in this way in the image plane are then projected back onto the selected plane in a 3D scene to obtain their shadows (dead zones).

Visualization of cropped objects with a common visibility area in the camera image plane is shown in Figure 3a. The walls and obstacles in the field of view are highlighted with a bold blue outline. Figure 3b shows visualization of calculated dead zones caused by obstacles on the floor and specific wall.

When determining the visibility polygon on the floor plane, we need to ensure that the objects getting into this polygon are visible in full size, i.e., with the desired upper bound visibility height, as shown in Figure 4. The camera’s field of view, taking into account the desired upper bound height, is determined by intersection of the visibility polygon at the upper bound height plane and the plane itself.

Thus, the described approach allows calculating the camera’s view area on any plane of the scene, considering dead zones, the required PPM level, and the desired upper bound visibility height (in the case of the floor plane). The latter aspect enables transferring all calculations to a two-dimensional space to calculate the metrics of the room coverage, as well as for the analysis of objects getting into any camera’s field of view. Thus, information about the shape of objects in the room is not lost, and the time and complexity of calculations are reduced.

### 3.3. Determining Highlights from Windows in the Image

The effect of sunlight in the camera image can be divided into two types: highlights and flares. Highlights (Figure 5) appear in the image in places with a lot of light, which in turn darkens the rest of the image. This does not allow the high-quality observation of certain objects, for instance, identifying people. Modern cameras can be equipped with the WDR [12] technology, which allows taking a series of frames with different exposure intervals and combining them into one image to achieve better image quality. However, such solutions are usually much more expensive than typical cameras.

Our method of automatic placement of visual sensors takes into account possible highlights from windows in case a window falls into the camera’s field of view and the camera does not have WDR. The horizontal angle between the camera and the nearest point of the window is an indicator of how high the probability of the highlight occurrence in the image is. The use of only a horizontal angle (without a vertical one) is due to the fact that the view area size is more affected by the camera’s pitch when installing. The model does not take into account the geographical location of the premises. Therefore, we use an idealized light pattern, when sunlight from the window can penetrate into the room at any horizontal angle.

To estimate probable highlight occurrence in the image, the following steps are performed:For each window in the room, its visible part is determined (the part of window which falls into camera’s field of view).If some part of the window is visible, then the horizontal angle between the camera pointing direction and the line from the camera to the nearest point of the window is calculated, as shown in Figure 6. The wider this angle, the higher the chance of highlights and flare appearing in the image.When the angle α is determined, the probability of the image highlight for each camera is calculated:
(4)$$p=\min\left(1,\overset{m}{\underset{i=1}{\sum }}\;k^{i}\left(1-\frac{2\alpha^{i}}{fov_{h}}\right)\right),$$
where $\alpha^{i}$ is the angle of incidence of light on the camera lens; $k^{i}$ is the intensity of the *i*-th light source; ${\mathrm{fov}}_{h}$ is the camera’s horizontal angle of view; and *m* is the number of light sources.

Thus, if some part of the window is visible in the camera image, then with an increase in the angle between the camera direction and the nearest point of the window, the penalty of possible highlights in the objective function will decrease. The chance of highlights occurrence is normalized in the range from 0 to 1.

### 3.4. Optimizing Camera Placement

To find the best possible solution, the task of automatic camera placement should be represented as a set of objective functions for a multi-objective optimization problem. For this, we propose two objective functions: minimizing the equipment cost and maximizing the room coverage with specified PPM requirements and room features. The solution is also imposed with restrictions on the camera placement coordinates and the equipment maximum total cost.

The objective function of the equipment cost minimization calculates the total cost of all cameras:(5)$$F_{c}=\overset{n}{\underset{i=1}{\sum }}Cost^{i},$$
where *n* is the number of cameras when placing and *Cost^i^* is the cost of the *i*-th camera.

The general objective function of the room coverage allows placing cameras while maximizing the coverage of doors, RoIs, the local area of cameras, and the overall area of the room coverage. As input, the function takes pre-calculated data of room model and camera parameters generated by the genetic algorithm. The height of the cameras is set in advance for the walls and ceiling, respectively. The camera is considered to be placed on the wall if its position enters the placement area along the room contour, otherwise it is placed on the ceiling. In addition, the roll angle of the cameras is set to 0 degrees.

The genetic algorithm generates 5 parameters for each camera (Table 3).

The general objective function of the room coverage consists of three subfunctions, which will be discussed further. The obtained results of the subfunctions are summarized in the end. The final value of the function is normalized in the range [0, 1], where the higher the value, the better the coverage of the room.

#### 3.4.1. Maximizing the Camera Coverage Area

The task of this subfunction is to maximize the total coverage area of the room and maximize the local coverage areas of each camera. The camera coverage area maximization subfunction is based on the weight function:(6)$$F_{s}=w_{o}\frac{S_{o}}{S_{r}}+w_{l}\frac{\sum_{i=1}^{n}\left(1-p^{i}\right)\frac{S^{i}}{S_{r}}}{n},$$
where $S_{o}$ is the overall coverage area of the room with cameras; $S_{r}$ is the area of the room; $S^{i}$ is the coverage area of the *i*-th camera; *n* is the number of cameras; $p^{i}$ is the probability of highlight occurrence for the *i*-th camera; $w_{o}$ is the weight to cover the overall area of the room; and $w_{l}$ is the weight to cover the local areas of the cameras. The function value is normalized in the range [0, $w_{o}$ + $w_{l}$ = 1].

#### 3.4.2. Maximizing the RoI Coverage

The main task of this subfunction is to maximize the coverage of regions of interest when placing cameras. For each zone the maximum coverage relative to each camera, but not the total for all, is calculated:(7)$$F_{roi}=w_{roi}\overset{m}{\underset{j=1}{\sum }}\underset{1\leq i\leq n}{max}\left(C^{ji}\left(1-p^{i}\right)\right)\frac{S_{roi}^{j}}{\sum_{k=1}^{m}S_{roi}^{k}},$$
where m is the number of RoI; n is the number of cameras; $C^{ji}$ is the percentage of the *j*-th RoI coverage by the *i*-th camera; $p^{i}$ is the probability of the highlight occurrence in the *i*-th camera; $S_{\mathrm{roi}}^{j}$ is the area of the *j*-th RoI; and $w_{\mathrm{roi}}$ is the constant coefficient for regions of interest coverage. Thus, the value of the function will always be normalized in the range [0, $w_{\mathrm{roi}}$].

#### 3.4.3. Maximizing the Door Coverage

The main objective of this function is to maximize the door zones coverage with the target PPM and to minimize the horizontal and vertical angles between the camera and the door’s normal. If the door opens inside the room, then the camera should be placed on the side of the door handle to minimize the possibility of creating a dead zone by the door itself when it is opened. In addition, main doors coverage is in higher priority relative to secondary doors.

The door coverage area maximization subfunction is based on the weight function:(8)$$\left\{ \begin{array}{c} F_{D}=\frac{k_{m}\cdot \sum_{i=1}^{m}\left(\underset{F_{mroi}^{ik},1\leq k\leq r}{argmax}\left(F_{mroi}^{ik}+F_{ma}^{ik}\right)\right)+k_{s}\cdot \sum_{j=1}^{n}\left(\underset{1\leq k\leq r}{max}\left(F_{sroi}^{jk}\right)\right)}{m\cdot k_{m}+n\cdot k_{s}}\\ \frac{+\sum_{i=1}^{m}\left(\underset{F_{ma}^{ik},1\leq k\leq r}{argmax}\left(F_{mroi}^{ik}+F_{ma}^{ik}\right)\right)}{m}\\ F_{sroi}^{jk}=w_{d}\cdot \left(1-p^{k}\right)\cdot \frac{S_{sroi}^{jk}}{S_{sroi}^{j}}\\ F_{mroi}^{ik}=w_{d}\cdot \left(1-p^{k}\right)\cdot \frac{S_{mroi}^{ik}}{S_{mroi}^{i}}\\ F_{ma}^{ik}=\left(1-p^{k}\right)\cdot \left(\frac{w_{\alpha }}{sign\left(\alpha^{ik}\right)}\cdot \left(1-\frac{\left|\alpha^{ik}\right|}{90}\right)+w_{\beta }\cdot \left(1-\frac{\left|\beta^{ik}\right|}{90}\right)\right)\\ sign\left(\alpha \right)=\{\begin{array}{c} 1,\alpha >0\\ 2,\alpha \leq 0 \end{array} \end{array},\right.$$
where m is the number of main doors; r is the number of cameras; $S_{\mathrm{mroi}}^{\mathrm{ik}}$ is the coverage area of the RoI of the *i*-th main door with the *k*-th camera; $p^{k}$ is the probability of the highlight occurrence in the *k*-th camera; $S_{\mathrm{mroi}}^{i}$ is the area of the *i*-th main door RoI; $w_{d}$ is the constant coefficient for covering the RoI of all types of doors in the room; $k_{m}$ is the weight for main doors; n is the number of secondary doors; $S_{\mathrm{sroi}}^{\mathrm{jk}}$ is the coverage area of the RoI of the *j*-th secondary door with the *k*-th camera; $S_{\mathrm{sroi}}^{j}$ is the area of the *j*-th secondary door RoI; $k_{s}$ is the weight for secondary doors; $\alpha^{\mathrm{ik}}$ is the horizontal angle between the *k*-th camera and the *i*-th door in the range [0, 90]; $w_{\alpha }$ is the constant coefficient for the horizontal angle of the camera; sign(α) is the sign of the horizontal angle; $\beta^{\mathrm{ik}}$ is the vertical angle between the *k*-th camera and the *i*-th door in the range [0, 90]; and $w_{\beta }$ is the constant coefficient for the camera vertical angle.

The horizontal angle α can vary in the range [−180, 180] and is considered positive if the camera is on the side of the door handle, otherwise it is negative. If the door opens inside the room and the camera is not installed on the side of the door handle, then the constant coefficient for the horizontal angle is divided by two, thereby imposing a penalty for non-optimal placement relative to the door.

A restriction is also imposed on the horizontal angle between the camera and the door: if α is in the range from 0 to a given angle, it is assumed that it is equal to 0. This reduces the penalty when the camera turns aside from the handle side of the door, as shown in Figure 7. If the door opens to the outside of the room, then this range extends to the opposite side of the door relative to the door handle.

Weights $k_{m}$ and $k_{s}$ are introduced in order to take into account a weighted average when calculating metrics for doors of various types. Thus, the value of the function will always be normalized in the range [0, $w_{d}+w_{\alpha }+w_{\beta }$].

#### 3.4.4. Choosing the Pareto front Optimal Solution

To obtain an optimal solution that best corresponds to the quality/cost ratio, the Pareto front is formed at the output of multi-objective optimization in the genetic algorithm [13]. At the same time, the metric for the quality criterion should be above the specified threshold value (in our case, we empirically chose 0.8). Furthermore, an important factor is that the optimal solution should not be the most expensive one.

To perform this multi-criteria decision making from the calculated Pareto front, we use the TOPSIS [14] method. Experimentally, it was found that solutions with TOPSIS weights (0.8; 0.2) and (0.9; 0.1) are often the most advantageous in terms of the quality/cost ratio because subsequent solutions do not demonstrate a significant increase in the room coverage function value, taking into account the increase in cost. Therefore, it was decided that the point with weights (0.8; 0.2) is the optimal solution in terms of quality/cost ratio, and the point with weights (0.9; 0.1) is the more expensive optimal solution (Figure 8).

## 4. Results

To conduct experiments based on real premises, more than 50 scenes of various types were modeled: open space offices, corridors, industrial premises, entrances, dining rooms, halls, and terraces, examples of which can be seen in Figure 9. During experiments, we considered fundamentally different conditions: premises with both large open spaces and narrow corridors; different number and placement of obstacles, doors, RoI; and various features of window placement, etc.

As for the experiments themselves, by placing a different number of cameras in a room, we can determine the effect of their number on the value of the room coverage function at given coverage priorities. In addition, the experiments are aimed at assessing the correctness of the cameras’ placement, taking into account the light from windows, where it is expected that cameras without the WDR will be located on the side of windows, i.e., in the places most protected from direct sunlight.

For the experiments, typical camera models were taken. Table 4 shows their parameters. Table 5 shows the parameters of the premises modeled in the experiments. The coefficients described in Table 6 are set for the general function of the room coverage. Having set such coefficients, we place the emphasis on covering doors and RoIs in the room.

Based on the simulated premises, the optimal parameters of the genetic algorithm, as well as the steps of these parameters, are identified. The solution of multi-objective optimization is carried out via the gamultiobj Pareto front search function implemented in Matlab. The parameters applied when testing the function used are presented in Table 7.

A combination of a population of 1024 and 64 generations most commonly has the best optimal solution and reaches a higher average value of the global Pareto front room coverage function faster throughout all iterations with a fixed number of optimization function launches per camera equal to 65,536. Populations of 512 and smaller for one camera are insufficient because the criteria space is large enough, which requires a large population in the first place. By increasing the number of generations, we allow the genetic algorithm to find a more optimal solution, but it will take longer time, which may not justify itself. Table 8 shows the results of comparing the combination of population size and the number of generations.

The grid step for placing cameras is 0.25 m, and the steps of pitch and yaw of the camera are 2 degrees. If the values are taken too small, then the dimension of the criteria space grows, which is why, with the same number of generations and the size of the genetic algorithm population, it does not have time to converge to the optimal value. During the experiments, it has been found out that the pitch and yaw of the camera do not affect the convergence of the genetic algorithm as much as the grid pitch. With chosen values, the best results were obtained for the majority of premises.

### 4.1. Optimizing Camera Placement

Let us determine the influence of a different number of cameras on the value of the room coverage function. If, with an increase in the number of cameras, the value of the function will grow slightly or not at all while the cost of the equipment grows, then the optimal number will be considered the one that causes significant function growth, taking into account the increase in cost. Let us consider the results of testing on two premises with one to five cameras installed, while comparing the optimal solutions. In addition, for a better understanding of the obtained value of the room coverage general function, we consider the values of each of its subfunctions (Figure 10 and Figure 11).

The first room (Figure 12a) has a complex geometric shape, five doors (including two main doors), and one large RoI. The room has a lot of bends, that is why, in our opinion, it takes two or three cameras to cover it.

Figure 10 shows a comparison diagram of the different number of cameras for the first room. It is apparent that one camera is enough to fully cover the RoI, and one door also falls into the camera’s view area, as shown in Figure 12b. To cover all the doors in the room, two cameras are enough, also with a covered region of interest. However, two cameras do not allow coverage of the entire room (Figure 12c). If the entire area of the room needs to be covered, then three cameras will be the optimal solution because with a larger number the equipment cost grows significantly whereas the coverage area of the room increases slightly.

The second room (Figure 13a) is an open space office with a large number of obstacles and windows. The room is elongated and has two partitions that do not allow capturing the entire area with the required PPM level with only one camera. Therefore, we believe that it also takes two or three cameras for its optimal coverage.

Figure 11 shows a comparison diagram of the different number of cameras for the second room. Here, one camera is sufficient to cover both doors because they are located next to each other, as shown in Figure 13b. Adding a second camera provides an opportunity to almost completely cover all the RoIs (Figure 13c). To increase the coverage area of the entire room, three cameras are enough (Figure 13d) because a larger number does not give a significant increase considering the increase in the equipment cost.

Thus, the testing shows that the different number of cameras installed leads to different values of the total coverage of the room, and not necessarily a larger number of cameras will give an increase in any subfunction or general function of the room coverage. For the rooms used in testing, two cameras were enough to cover all the doors and mandatory areas, and most of the total area of the room. To completely cover the entire room, three cameras are most commonly enough for this type of room.

### 4.2. Assessment of Camera Placement Considering Highlights from Windows

The purpose of this test is to assess the method of camera placement, taking into account the light from the windows. The test has been conducted in a second room with two cameras. A comparison for the Pareto front optimal solutions is provided.

The camera placement without taking into account the highlights is shown in Figure 14a. The value of the room coverage function is 0.88 at a cost of USD 320. It can be observed that the first camera covers both doors and part of the RoI. At the same time, a part of the window falls into the camera’s field of view, as shown in Figure 14b. The second camera is placed in the corner of the room in such a way as to increase the coverage area of the RoIs and the total area of the room. In this case, all windows fall into the camera’s field of view (Figure 14c). In the described case, the use of similar cameras but with WDR will increase the cost of the solution up to USD 410.

At the same time, the proposed method allows us to find another solution, without increase in cost. Figure 15a shows the optimal location of cameras without WDR with the window detection enabled. The value of the coverage function is lower than without the window detection and is equal to 0.78 with a total cost of equipment of USD 200. The first camera also covers both doors but captures a smaller part of the RoI because windows should not fall into the camera’s field of view, so that a penalty for highlights is not included (Figure 15b). The second camera is placed in the opposite corner of the room in relation to the previous experiment in order to minimize the possibility of the highlight occurrence, while covering most of the RoIs and the total area of the room (Figure 15c).

Figure 16a shows the optimal solution for the camera automatic placement, taking into account the windows for cameras with and without the WDR function. The value of the room coverage function for this solution is 0.85 at a cost of USD 276, which is only 0.03 less than when placing cameras without taking into account windows. When comparing the same-cost solutions, the function value is 0.87 at a cost of USD 324, which is 0.01 less than the value without taking into account windows. However, this solution includes the use of two cameras with WDR, which makes the cost higher.

The compromise solution can be the use of two different cameras (with and without WDR). The camera without WDR still covers both doors and a part of the RoI without the window falling into the camera’s field of view (Figure 16b). The second camera with WDR is placed opposite the windows and covers both RoIs completely (Figure 16c). The characteristics of the solutions described above are compared in Table 9.

Thus, cameras without WDR are installed in the corners of the room in such a way as to minimize the window falling into the field of view. The cameras with WDR are more expensive than the analogues. Therefore, placing cheaper cameras with similar characteristics without compromising image quality due to the absence of windows in the field of view appears an excellent solution to reduce costs.

### 4.3. Performance Estimation

Let us consider the performance of our solution for three rooms of different complexity (Figure 17). We modeled 3 rooms of different shapes, different numbers of RoIs, obstacles, doors, and windows.

In a simple case, the room is presented by two adjacent chambers connected by a narrow passage. It has 2 doors, 4 windows located along one wall, and two RoIs. We tried to model such conditions so that the intuitive solution was not obvious at first glance. At the same time, the task is considered easier than the cases b and c for the algorithm, because the rooms have more features. So, in the “medium” case, there are more doors and obstacles of both types. Additionally, the “hard” case is represented by a long corridor with many bends, doors, obstacles, and non-coplanar windows. The algorithm performance is shown in Table 10.

It can be seen that for all cases the cameras were placed in such a way as to maximize all types of coverage and minimize the cost. In addition, the obtained metrics for each subfunction fully correspond to the priorities that were set for the algorithm. First of all, doors and RoI are covered, which give the greatest increase in the value of the function, then the entire area of the room, and the local view areas of each camera.

At first glance, the computing time of the algorithm may seem large. However, it should be borne in mind that when using genetic algorithms, there is a search over a rather large space of parameters to take into account all the features of the model. The performance time of our solution is comparable to similar works, e.g., [5,6]. In addition, the algorithm and the models were implemented as a prototype using Matlab environment, which, as a rule, shows worse performance results than the frameworks used to write the final product. The computation time can be reduced by simplifying the model, for example, by changing the grid step, setting preferable areas for camera placement [11], and using second-order optimization, e.g., [9,10].

## 5. Discussion

Ensuring requirements for equipping a smart room is a multi-parameter task that can be decomposed and partially automated within known limitations. So, despite the different types of smart spaces, some features must be observed in any case. When talking about equipping a smart space with a set of sensors, we should provide the entire area coverage regardless of the type of premise in question. In addition, when it comes to sensory equipment, the necessary signal level must be provided for every sensor. At the same time, where there is no need for high resolution, it makes sense to use cheaper equipment.

In our example, for simplicity and clarity, we considered the placement of CCTV cameras because sensors of this type have known difficulties: we should avoid glare from windows and provide the required image quality for known monitoring and recognition tasks. This allowed us to simulate a real-life use case at the same time without reference to the features of a particular room. The effectiveness of the developed method was tested on the basis of simulation results, in which real rooms of various types were generated.

The proposed method is flexible due to the assignment of weights to certain aspects during the arrangement. We can both focus on maximizing the coverage of the total area of the room and on covering all doors from a certain angle. The weight function can be fitted in a such way to allow one to find a satisfactory solution for different customer requirements.

With minor modifications, the proposed method can be scaled to other types of sensors and devices (infrared and multispectral cameras, ultrasonic sensors, microphones, dynamics, etc.) with specified characteristics of operating range and field of view. These features make our solution applicable for automating the tasks of building modeling and intelligent systems design (smart spaces, smart factories, specialized cyber-physical systems, etc. [15,16,17]).

## 6. Conclusions

The present work considers new aspects that were not previously taken into account when solving the problem of the automatic placement of visual sensors in the room. Thus, our method considers the developed representation of the doors to ensure an acceptable image quality for recognizing incoming people. It also considers specified areas of interest in the model to ensure the required image quality (PPM level) for any highly specialized tasks of computer vision (such as object detection and recognition).

In addition, our method is intended to minimize the total cost of the equipment used. By accounting windows in the model, we solve the problem of dealing with high-contrast lighting conditions because cameras without WDR are cheaper. This allows finding acceptable solutions with the limited budget.

Further research can be carried out in several directions. First of all, we focus on improving the mathematical model of the sensor, to take into account lens distortions for various types of cameras. This will allow calculating the visual sensor field of view with even greater accuracy. Other sensor types can be added to the model as well. As for the room model, it is possible to add prioritization for areas of interest, so that when placing a small number of cameras, the areas with the highest priority are covered first. Highlights in the image can occur not only from natural light sources, but also from artificial ones. In this regard, we plan to add lamps and other light sources to the model. Finally, it is worth adding the feature of determining the optimal height of the camera position when placing it. This will increase the parameter space when optimizing, but also allow more flexible method settings.

## Figures and Tables

**Figure 1 sensors-22-07806-f001:**
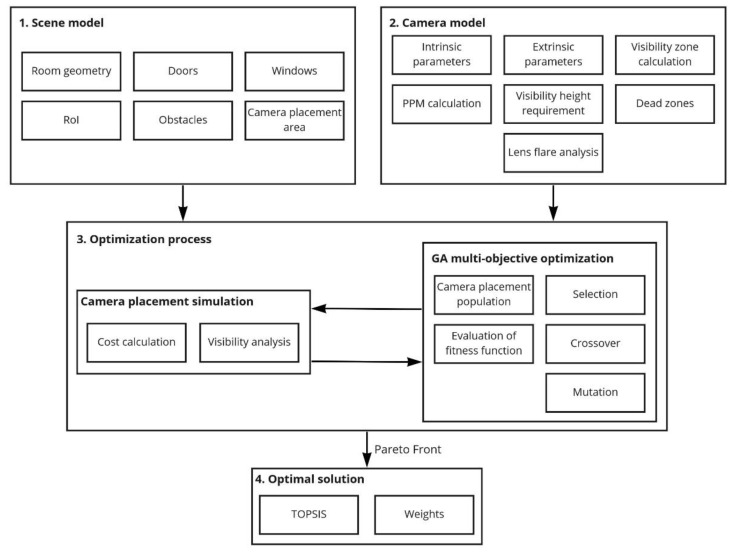
Proposed method of automatic placement of visual sensors.

**Figure 2 sensors-22-07806-f002:**
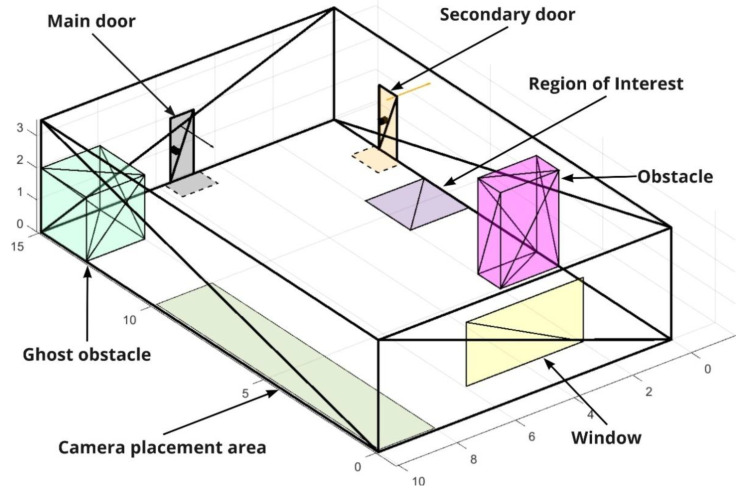
Room model.

**Figure 3 sensors-22-07806-f003:**
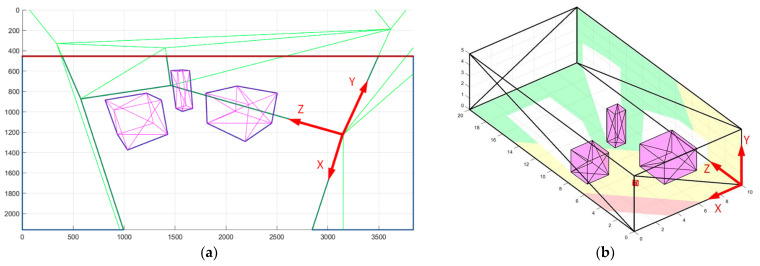
Visual sensor modeling: (**a**) camera view horizon (red); and (**b**) dead zones caused by obstacles (white).

**Figure 4 sensors-22-07806-f004:**
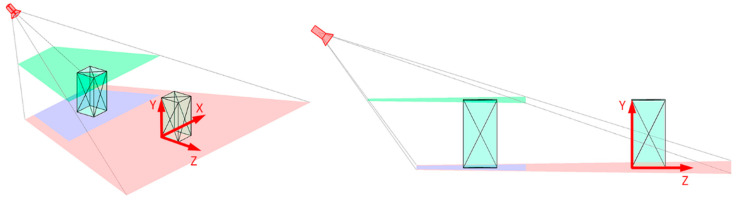
The view area upper bound height.

**Figure 5 sensors-22-07806-f005:**
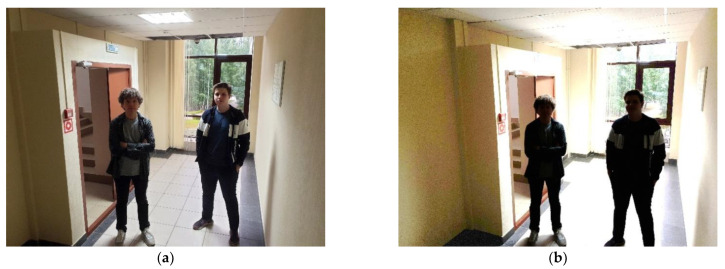
Highlights in the camera image: (**a**) WDR on; (**b**) WDR off.

**Figure 6 sensors-22-07806-f006:**
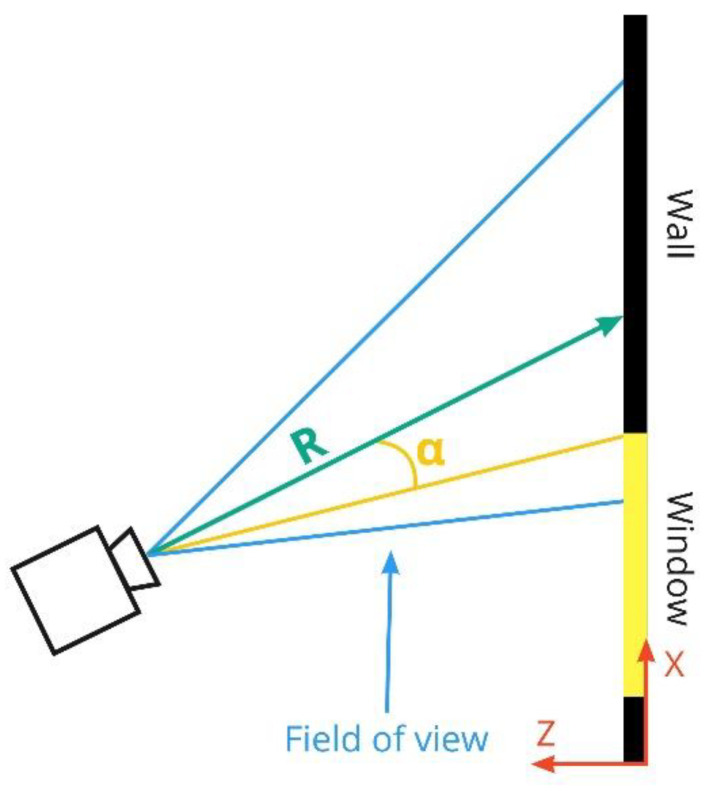
Modeling the calculation of camera highlights.

**Figure 7 sensors-22-07806-f007:**
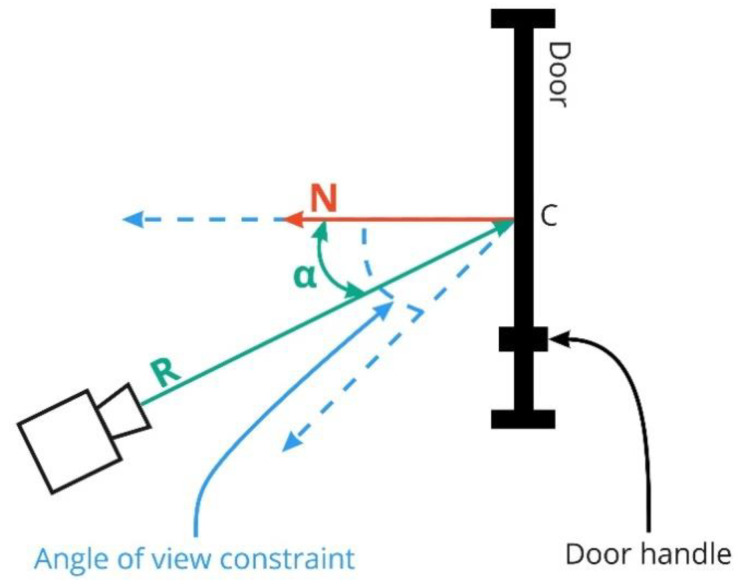
Covering doors.

**Figure 8 sensors-22-07806-f008:**
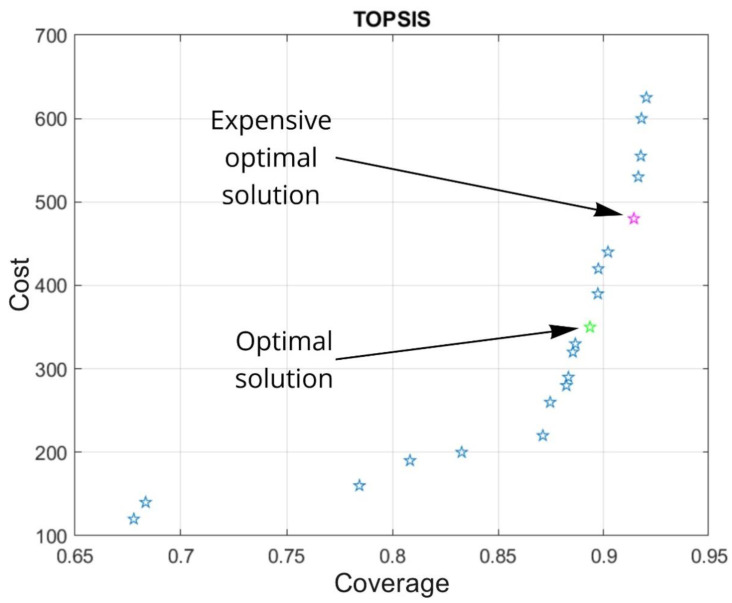
Pareto-front solutions for camera placement. Blue stars—optimal Pareto solutions, green—optimal TOPSIS solution, pink—expensive optimal TOPSIS solution.

**Figure 9 sensors-22-07806-f009:**
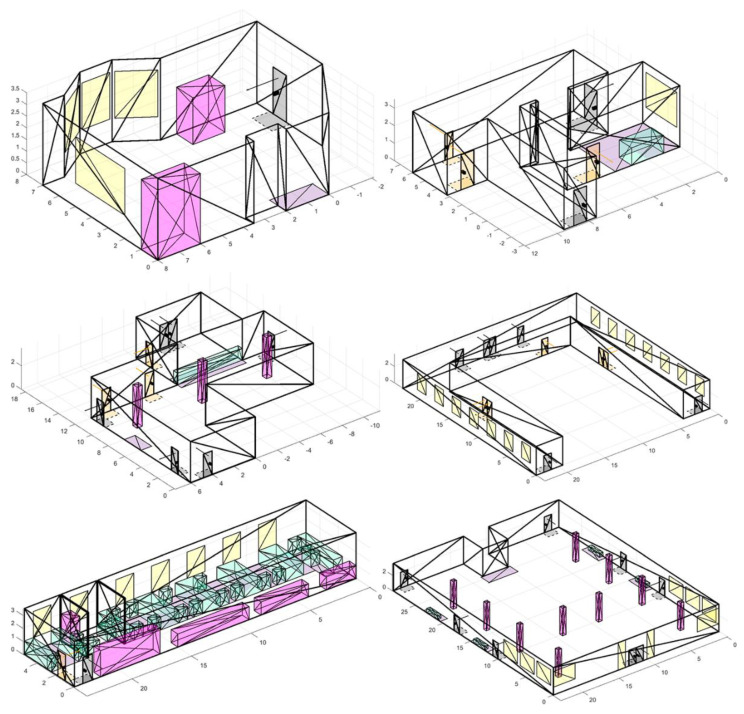
Premises models examples.

**Figure 10 sensors-22-07806-f010:**
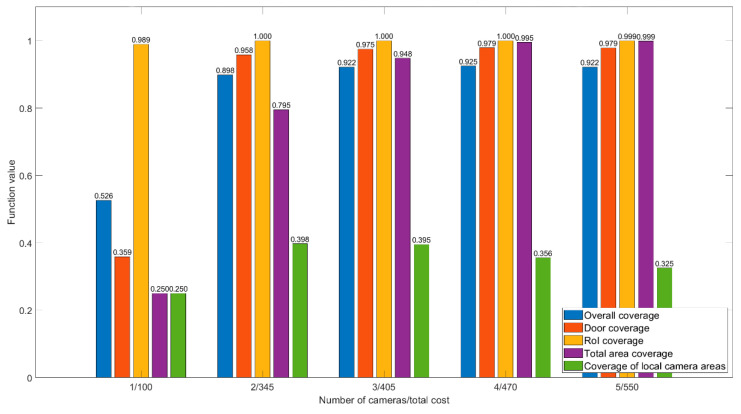
Function values for different number of cameras, room 1.

**Figure 11 sensors-22-07806-f011:**
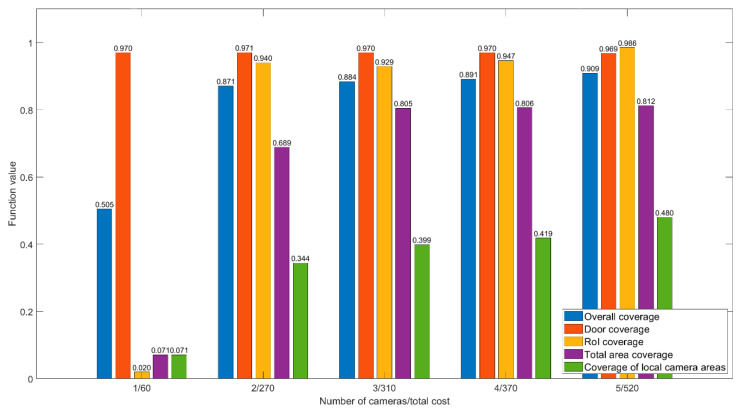
Function values for different number of cameras, room 2.

**Figure 12 sensors-22-07806-f012:**
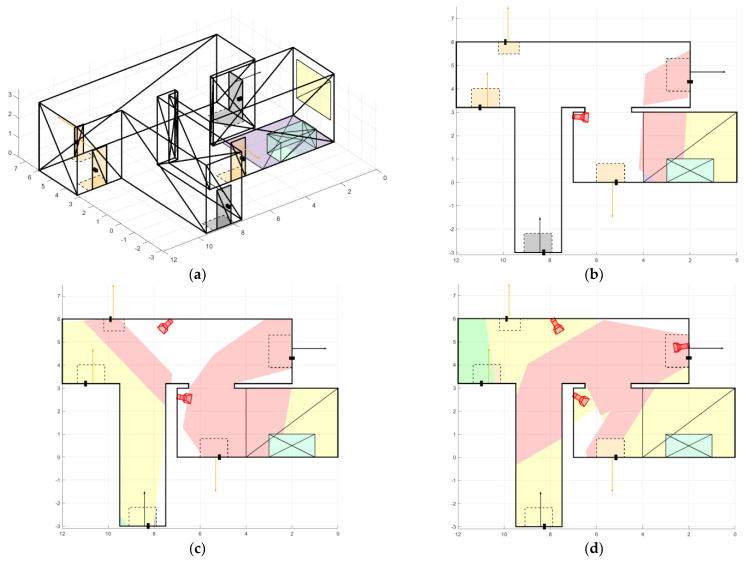
Placement of cameras in room 1: (**a**) room 1 model; (**b**) single camera placement; (**c**) placement of two cameras; and (**d**) placement of three cameras.

**Figure 13 sensors-22-07806-f013:**
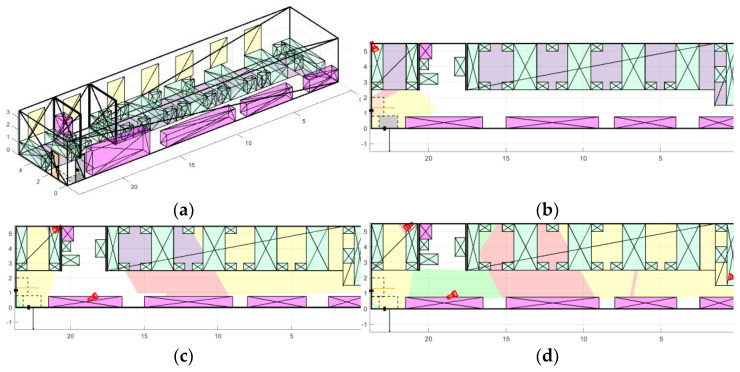
Placement of cameras in room 2: (**a**) room 2 model; (**b**) single camera placement; (**c**) placement of two cameras; and (**d**) placement of three cameras.

**Figure 14 sensors-22-07806-f014:**
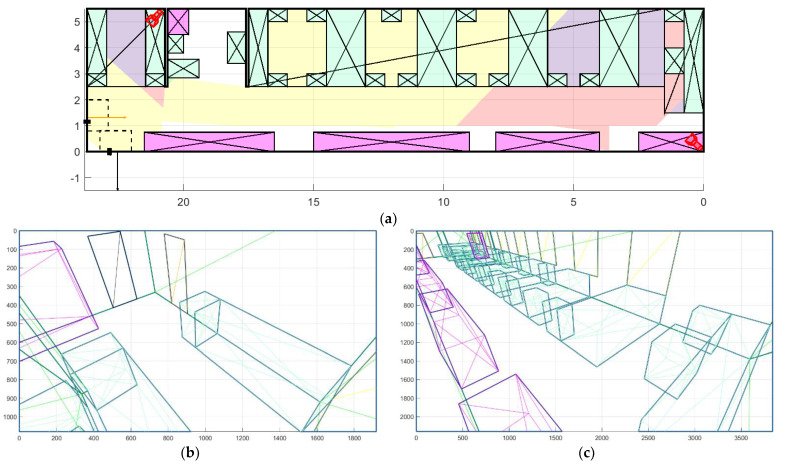
Placement of cameras without taking into account the windows in the room 2: (**a**) floor plan; (**b**) view from the first camera; and (**c**) view from the second camera.

**Figure 15 sensors-22-07806-f015:**
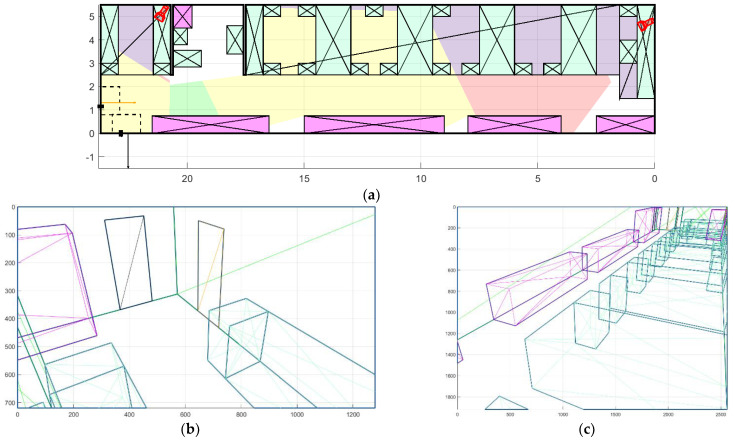
Placement of cameras without WDR, taking into account the windows in the room 2: (**a**) floor plan; (**b**) view from the first camera; and (**c**) view from the second camera.

**Figure 16 sensors-22-07806-f016:**
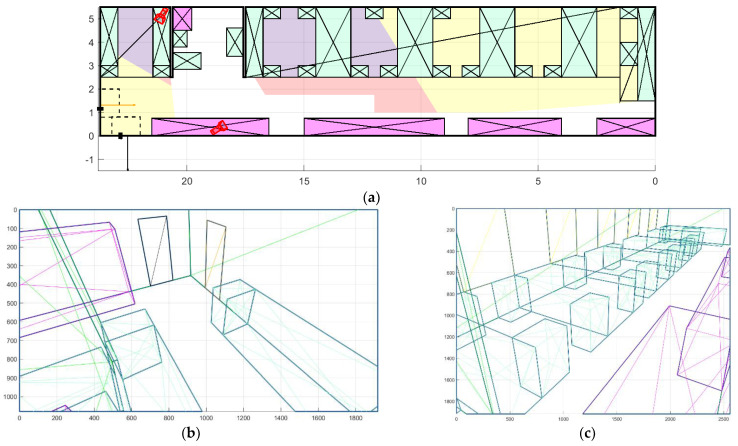
Placement of cameras with and without WDR, taking into account highlights in the room 2: (**a**) floor plan; (**b**) view from the first camera without WDR; (**c**) view from the second camera with WDR.

**Figure 17 sensors-22-07806-f017:**
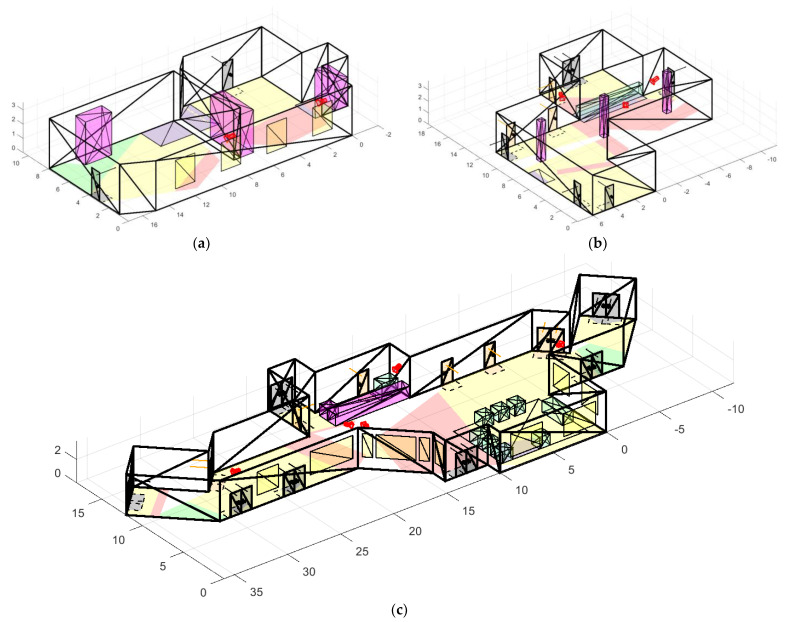
Found solutions for rooms of varying complexity: (**a**) simple case; (**b**) medium case; and (**c**) hard case.

**Table 1 sensors-22-07806-t001:** Methods for solving the problem of automatic camera placement.

Autor, Publication Date	Technique	Task	Disadvantages
Horster and Lienhart, 2006 [2]	2D	Determine the minimum number of visual sensors of a certain type as well as their positions and poses in the space such that coverage is achieved. Given different types of visual sensors determine how to obtain coverage while minimizing the total cost of the sensor array.	The specified area can only be a rectangle. The scene does not have obstacles, doors, or other obstructions. The 2D method is not suitable for accurate coverage calculation. The position of light sources is not taken into account.
Abdalla and Asirvadam, 2017 [3]	2D	Multi-camera coverage modeling setup prior to the placement optimization to support offline sensor planning in surveillance systems.	The scene does not have obstacles, doors, or other obstructions. Minimizing the cost of cameras when placing is not provided. The 2D method is not suitable for accurate coverage calculation. The position of light sources is not taken into account.
Van den Hengel and Hill, 2009 [4]	3D	Identify the optimal placements for a set of cameras given the shape of the space to be observed, the number and type of cameras available, and the level of automated software assistance desired.	Doors, windows, and regions of interest are not considered. Minimizing the cost of cameras when placing is not provided. The room model does not take into account the obstacles forming dead zones. The upper bound height of the camera view area is not taken into account. The position of light sources is not taken into account.
Kim and Ham, 2019 [5]	2D	Determine the optimal number, types, locations, and orientations of cameras that maximize visible coverage and minimize total costs of a camera network given jobsite-specific constraints (e.g., limited budgets, installable locations).	Questionable solution on considering obstacles in 2D mode. The 2D method is not suitable for accurate coverage calculation. The position of light sources is not taken into account.
Sourav and Peschel, 2022 [6]	BIM	Maximize the multi-camera coverage while minimizing budget.	Doorways are not taken into account. Option to set a specific area for placing cameras is not available. The position of light sources is not taken into account.
Amiri and Rohani, 2014 [7]	3D	A camera toolset in SketchUp and a decision support system using an enhanced particle swarm optimization algorithm. The objective for the proposed algorithm was to have a good computational performance to quickly generate a solution for the automatic camera placement problem.	Minimizing the cost of cameras when placing is not provided. The room model does not take into account the obstacles forming dead zones. The position of light sources is not taken into account.
Albahri and Hammad, 2017 [8]	BIM	Find the optimum cameras’ type, number, and placement inside a building that can offer the maximum camera coverage with the minimum cost.	Doorways are not taken into account. The position of light sources is not taken into account.

**Table 2 sensors-22-07806-t002:** Required PPM values for different tasks.

Task	Minimal PPM Level Required
Identification	250
Recognition	125
Observation	62
Detection	25
Monitoring	12

**Table 3 sensors-22-07806-t003:** Parameters generated by the genetic algorithm.

Parameter	Range	Step	Type
X	from the minimum to the maximum point of the room on the X axis (room width)	0.25 m	Integer
Z	from the minimum to the maximum point of the room on the Z axis (room length)	0.25 m	Integer
Pitch	from −90 to 0 degrees	2 degrees	Integer
Yaw	from −180 to 180 degrees	2 degrees	Integer
Camera model: FOV, image resolution, WDR, price	-	1	Integer

**Table 4 sensors-22-07806-t004:** Camera parameters in experiments.

Image Resolution, Pixels	Angle of View, Degrees	WDR Availability	Cost, USD
1280 × 720–3840 × 2160	40–110	true, false	40–270

**Table 5 sensors-22-07806-t005:** Premises parameters in experiments.

Parameter	Value
Camera placement height	3–4 m (depends on the room)
The maximum desired height of the camera view area	2 m
Maximum equipment cost limitation	-
Placement on the wall with indentation	0.1–0.3 m
Number of doors	1–10
Area of the room filled with obstacles	0–30%
Number of windows	0–15

**Table 6 sensors-22-07806-t006:** Coefficients of the room coverage function.

Coefficient Description	Coefficient	Value
Coverage of doors zones	$w_{d}$	0.3
Door’s horizontal view angle	$w_{\alpha }$	0.1
Door’s vertical view angle	$w_{\beta }$	0.1
Region of interest	$w_{roi}$	0.3
Overall area of the room coverage	$w_{o}$	0.1
Local camera areas coverage	$w_{l}$	0.1

**Table 7 sensors-22-07806-t007:** Testing parameters used for *gamultiobj*.

Parameter	Description	Value
CreationFcn	Used to create an initial population	gacreationsobol
CrossoverFcn	Crossover type	crossoversinglepoint
MutationFcn	Mutation type	mutationpositivebasis
MaxGenerations	The maximum number of iterations before the algorithm stops	64
PopulationSize	Population size	1024

**Table 8 sensors-22-07806-t008:** Comparison of the combination of population size and number of generations for 2 cameras.

Population Size/Number of Generations	The Average Value of the Room Coverage Function of the Global Pareto Front	Number of the Optimization Function Runs for CONVERGENCE, Approx	Total Number of the Optimization Function Runs, Approx
256/256	0.7	56,000	58,000
384/170	0.72	50,000	78,000
512/128	0.72	50,000	106,000
768/85	0.77	120,000	130,000
1024/64	0.77	62,000	130,000
2048/32	0.77	123,000	130,000

**Table 9 sensors-22-07806-t009:** The comparison of the optimal solutions found.

No	Number of Cameras	Room Coverage Function	Cost	Types of Cameras	Visualization
1	2	0.88	USD 320	1. Without WDR 2. Without WDR Windows detection disabled	Figure 14
2	2	0.78	USD 200	1. Without WDR 2. Without WDR Windows detection enabled	Figure 15
3	2	0.85	USD 276	1. Without WDR 2. With WDR Windows detection enabled	Figure 16

**Table 10 sensors-22-07806-t010:** Comparison of solutions for rooms of varying complexity.

Case	Number of Cameras	Total Cost	Overall Coverage	Door Coverage	RoI Coverage	Total Area Coverage	Average Coverage of Local Camera View Areas	Time, s
Simple	2	USD 290	0.88	0.981	0.94	0.717	0.358	610
Medium	3	USD 320	0.892	0.954	1	0.851	0.302	1538
Hard	5	USD 402	0.808	0.848	0.941	0.844	0.173	8822

## Data Availability

Not applicable.

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
