# Peer review of "Automatic Placement of Visual Sensors in a Smart Space to Ensure Required PPM Level in Specified Regions of Interest"

_sensors, 2022, doi:10.3390/s22207806_

Round 1

Reviewer 1 Report

The article covers the problem of automatic placement of visual sensors, aiming at acceptable levels of placement (PPM) for specified regions of interest. Authors propose a cost-effective solution to address this problem, which has a lot of practical applications.

The problem of visual sensors (cameras) placement is not new. However, there are still some challenges to be overcome and the authors seem to follow one of such directions. The problem of efficient placement of cameras within a 3D room is tackled through a multiobjective algorithm described in this work (formulations).

Overall, the paper is quite good and easy to follow. English is correct and sentences are clear. However, I believe there are some issues to be considered before publication.

- Introduction section should be split off into two different sections. Actually, authors inserted a lot of "related works" descriptions into the Introduction, which I believe makes the paper poorly structured in this sense. So, I suggest the creation of a proper Related Works section, moving Table I and descriptions of previous works to that new section

Doing so, authors sould better define the problem scope and the employed techniques in the Introduction. With the new proposed organisation of the paper, Introduction should better guide the readers, and maybe 2 additional paragraphs slightly describing the problems and solutions would be worth it. 

The new Related Works section should better review the state-of-the-art, describing other works that comprise subjects such as Dependability, Energy efficiency and Mobility issues (just to cite a few), which are common in visual sensor networks and that are somehow related to the subject of this article. These are some sugestions to be cited and discussed:

- Coverage optimization of visual sensor networks for observing 3-D objects: survey and comparison. Xuebo Zhang, Boyu Zhang, Xiang Chen, Yongchun Fang. International Journal of Intelligent Robotics and Applications volume 3, pages 342–361, 2019. https://doi.org/10.1007/s41315-019-00102-6

- On Redundant Coverage Maximization in Wireless Visual Sensor Networks: Evolutionary Algorithms for Multi-objective Optimization. E. Rangel, Daniel G. Costa, A. Loula. Applied Soft Computing, 82, 105578, 2019. https://doi.org/10.1016/j.asoc.2019.105578

- Placement of Optical Sensors in 3D Terrain Using a Bacterial Evolutionary Algorithm. Szilárd Kovács, Balázs Bolemányi, János Botzheim. Sensors, 2022, 22(3), 1161. https://doi.org/10.3390/s22031161

Section 2, Materials and Methods, is a good piece of work!! 

Results are also consistent and properly discussed. I particularly liked the use of a Pareto front as a set of possible solutions, since there may be multiple solutions that fulfil the defined requirements.  

Author Response

Dear reviewer!

Thank you for the valuable comments! We have carefully reviewed the comments and have revised the manuscript accordingly. Our responses are given below.

  1. The Introduction section is divided into Introduction and Related Works. Both new sections have been revised and updated. We added new information about the topic in the Introduction and several state-of-the-art works discussing coverage optimization in visual sensor networks. We hope that the changes made will help the reader to better navigate the paper.
  2. As well, we added Discussion and updated Conclusion.
  3. In order to estimate performance of our approach, we added a subsection 4.3. Unfortunately, we cannot compare our approach with the well-known solutions due to difficulty in implementing ideal comparison conditions during the time allotted for revising the manuscript. However, we give the parameters of the algorithm performance for cases of different complexity. As can be seen from Table 10, the algorithm performance is compatible to the known solutions. Certainly, the time performance our approach has considerable potential for improvement. Note that we set ourselves the goal of developing a prototype to show the validity of our approach, and not a ready-made solution. Nevertheless, even at the prototype stage, our approach is comparable to known works in this area. We discuss this issue in the text.
  4. In addition, we proofread the text and made minor corrections to fix typos, to explain all the abbreviations, and to improve English. As well, we added the coordinate system in some figures for better understanding.

We hope you find the changes satisfactory. If not, we will gladly modify our manuscript according to additional recommendations.

Reviewer 2 Report

I have followinf concerns about the paper:

-A new sub-section can be created to include literature review, which can be named as Related Work. 

-Therefore, introduction part must be improved by adding new information about the topic.

-Equations may require citations, please check them.

-Include more up-to-date papers to the Related Work section.

-How about a performance comparison with a well-known approach.

-

Author Response

(The authors gave the same response as above.)

Reviewer 3 Report

The paper deals with  automatic placement of visual sensors in a smart space. The idea is interresting, presented ideas are clear. The paper is well written and self-explaining. I have a few recommendations:

- all abbreviations should be explained;

- some minor errors or typos in the text, please re-check the English;

- I really miss drawing of the coordinate system in some figures, e.g. Figure 3, Figure 4, Figure 7, so if possible, please add;

- the last part of the paper is Discussion, not Conclusion. I recommend to add the Conclusion part. 

- In the Discussion part I would like to ask the authors to introduce at least the easiest case and the hardest case from the mentioned 50 scenes, some comparison of the alrgorithm results. It would give a better overview of the performance.

- considering the algorithm, I haven't found some any information about the time consumption, at least for the presented cases. Please add, if possible. 

Author Response

(The authors gave the same response as above.)

Round 2

Reviewer 2 Report

Thanks for addressing my previous isssues and be carefull with the performance evaluation section in your future studies.